# Application of a Brain–Computer Interface System with Visual and Motor Feedback in Limb and Brain Functional Rehabilitation after Stroke: Case Report

**DOI:** 10.3390/brainsci12081083

**Published:** 2022-08-16

**Authors:** Wen Gao, Zhengzhe Cui, Yang Yu, Jing Mao, Jun Xu, Leilei Ji, Xiuli Kan, Xianshan Shen, Xueming Li, Shiqiang Zhu, Yongfeng Hong

**Affiliations:** 1Department of Rehabilitation Medicine, The Second Hospital of Anhui Medical University, No. 678 Furong Road, Economic and Technological Development Zone, Hefei 230601, China; 2Zhejiang Laboratory, Department of Intelligent Robot, Keji Avenue, Yuhang Zone, Hangzhou 311100, China; 3Ocean College, Zhejiang University, No. 866 Yuhangtang Road, Xihu Zone, Hangzhou 310030, China

**Keywords:** brain–computer interface, active rehabilitation training, stroke, motor function, brain function

## Abstract

(1) Objective: To investigate the feasibility, safety, and effectiveness of a brain–computer interface (BCI) system with visual and motor feedback in limb and brain function rehabilitation after stroke. (2) Methods: First, we recruited three hemiplegic stroke patients to perform rehabilitation training using a BCI system with visual and motor feedback for two consecutive days (four sessions) to verify the feasibility and safety of the system. Then, we recruited five other hemiplegic stroke patients for rehabilitation training (6 days a week, lasting for 12–14 days) using the same BCI system to verify the effectiveness. The mean and Cohen’s w were used to compare the changes in limb motor and brain functions before and after training. (3) Results: In the feasibility verification, the continuous motor state switching time (CMSST) of the three patients was 17.8 ± 21.0s, and the motor state percentages (MSPs) in the upper and lower limb training were 52.6 ± 25.7% and 72.4 ± 24.0%, respectively. The effective training revolutions (ETRs) per minute were 25.8 ± 13.0 for upper limb and 24.8 ± 6.4 for lower limb. There were no adverse events during the training process. Compared with the baseline, the motor function indices of the five patients were improved, including sitting balance ability, upper limb Fugel–Meyer assessment (FMA), lower limb FMA, 6 min walking distance, modified Barthel index, and root mean square (RMS) value of triceps surae, which increased by 0.4, 8.0, 5.4, 11.4, 7.0, and 0.9, respectively, and all had large effect sizes (Cohen’s w ≥ 0.5). The brain function indices of the five patients, including the amplitudes of the motor evoked potentials (MEP) on the non-lesion side and lesion side, increased by 3.6 and 3.7, respectively; the latency of MEP on the non-lesion side was shortened by 2.6 ms, and all had large effect sizes (Cohen’s w ≥ 0.5). (4) Conclusions: The BCI system with visual and motor feedback is applicable in active rehabilitation training of stroke patients with hemiplegia, and the pilot results show potential multidimensional benefits after a short course of treatment.

## 1. Introduction

Stroke is the second leading cause of death worldwide, but is the first leading cause of death in China, the world’s most populous country [1,2,3]. Stroke after interruption of cerebral perfusion causes rapid loss of brain function, often leading to hemiplegia [4], which describes partial or complete paralysis on one side of the body, usually due to extensive cerebral infarction contralateral to the main symptoms [5]. About 60% of stroke survivors experience persistent impairment of motor function and, consequently, need rehabilitation [6]. In addition, stroke survivors often suffer from a range of non-motor disabilities, including visual and cognitive impairment [7], the composite of which has a serious impact on daily activities. Therefore, effective treatment and rehabilitation of stroke patients have been a research focus for many years.

The treatment and rehabilitation of stroke patients entails extensive efforts to help the brain recover damaged neural connections and compensate for broken neuronal pathways. The conventional rehabilitative treatments target motor dysfunctions in the upper and lower limbs, and includes various facilitation techniques [8], functional electrical stimulation (FES) [9], and other interventions, which generally have lesser effects on the functioning of the upper limbs in a patient recovering from a serious stroke. In recent years, some emerging technologies have been applied to the rehabilitation of stroke, such as powered exoskeleton technology [10] and multi-degree-of-freedom stroke rehabilitation robot technology [11], and positive effects have been achieved. These emerging technologies may create new rehabilitation paradigms aimed at accelerating functional recovery in stroke patients [12]. 

Brain–computer interface (BCI) is a relatively new technology for exchanging information between the brain and external devices. With the help of BCI technology, patients can actively participate in rehabilitation training. BCI has been reported for the rehabilitation of stroke and similar disorders like Parkinson’s disease, cerebral palsy, and spinal cord injury, and can significantly promote the recovery of limb function in these patients [13,14,15,16]. However, how to get stroke patients to imagine target tasks more effectively and stably; how to monitor the performance of imaginary tasks; how to accurately capture and identify task-related EEG signals; how to convert EEG signals into views, movements, sounds, etc., in order to achieve the enhanced efficacy of BCI training with multiple sensory feedback; and how to solve BCI training fatigue to ensure effective training time are still challenges [17].

Most BCI rehabilitation systems adopt single feedback modality, such as systems based on vision [18,19,20], kinesthetic sense [21,22,23], or perception [24,25]. The main purpose is to provide perceptual or sensory feedback, and these feedback modalities can lead to motor function recovery. How these feedback patterns affect clinical outcomes remains elusive. However, a study has shown that sensory feedback of exercise may be a key factor in BCI-based rehabilitation, and single visual feedback is not sufficient to arouse functional gain [26]. In addition, immersive visual feedback may help enhance the effect of feedback [27,28].

In the present report, we introduce a new BCI system, namely the L-B300 Electroencephalogram (EEG) Acquisition and Rehabilitation Training System (Zhejiang Mailian Medical Technology Co., Ltd., Hangzhou, China). Compared with other BCI rehabilitation systems, the BCI rehabilitation system used in this report has the advantage of both visual and motor feedback and high time efficiency: its real-time feedback is only 110 ms, and this parameter is smaller than those of other systems (300 ms [29] and 200 ms [30]); it is so responsive and user-friendly that the user is not easy to feel tired [31]; it is very convenient in clinical use and can guide and timely monitor target task imagination of patients. The purpose of this study was to observe the feasibility and safety of this novel BCI system for rehabilitation of stroke patients with hemiplegia, and the effects of a short course of BCI treatment on patients’ limb movement and brain function.

## 2. Materials and Methods

Eight stroke patients with hemiplegia were recruited in the Department of Rehabilitation Medicine of the Second Hospital of Anhui Medical University to perform BCI rehabilitation training with visual and motor feedback from August 2021 to October 2021.

### 2.1. Subjects

Inclusion criteria: (1) 18–80 years old, either sex; (2) patients with de novo post-stroke hemiplegia were diagnosed by imaging examination as having had cerebral infarction or cerebral hemorrhage; (3) the strength of all the muscles of the upper and lower limbs on the hemiplegic side was grade 4 or less (manual muscle testing, MMT) [32], while there was no obvious dysfunction on the healthy side; (4) patients could sit without support for 30 min; (5) stable vital signs; (6) clear awareness and ability to participate in the rehabilitation intervention; (7) patients understood and signed informed consent.

Exclusion criteria: (1) patient’s condition continued to deteriorate, vital signs unstable; (2) had severe heart disease, or carried pacemakers that might interfere with the BCI; (3) patients with recurrent (≥2 times) cerebral infarction or cerebral hemorrhage; (4) poor cognitive level, with Mini-mental State Examination (MMSE) score less than 21, and, thus, unable to comply with rehabilitation therapy [33]; (5) patients receiving craniectomy; (6) patients after cranioplasty; (7) patients with motor dysfunction contralateral to the hemiplegic limbs; (8) could not sit alone for 30 min; (9) patients or their families refused to sign informed consent; (10) patients were participating in other clinical experiments.

General information of patients: three patients (48 ± 15 years old) were recruited for the initial feasibility and safety observation study. After verifying the feasibility and safety of the system, another five patients (48 ± 9 years old) were recruited for the effectiveness evaluation of the system. The demographic information of all eight patients is shown in Table 1.

This study was carried out in the Department of Rehabilitation Medicine of the Second Hospital of Anhui Medical University. The experimental method was approved by the Ethics Committee of the Second Hospital of Anhui Medical University [Approval No. YX2020-103(F1)] and implemented according to the ethical standards of the 1975 Helsinki Declaration (revised in 2008). Written informed consent was obtained from each participant before enrollment.

### 2.2. L-B300 EEG Acquisition and Rehabilitation Training System

The rehabilitation training system used in this study was the new L-B300 EEG Acquisition and Rehabilitation Training System. In this system, the patient wore an 8-lead EEG cap with electrodes over the left and right prefrontal cortex (FP1, FP2), left and right frontal cortex (F3, F4), left and right central area (C3, C4), frontal midline area (Fz), and central midline area (Cz), respectively. The electrode placement was in line with the 10/20 international standard lead system [34], as shown in Figure 1. The left and right ear clip electrodes (A1 and A2) were, respectively, the reference and bias electrodes. The patients’ bilateral upper/lower limbs were fixed on the corresponding left and right rotary shafts of the system for upper/lower limb rehabilitation training. The system provides a training mode, in which the equipment was driven completely by the patient’s EEG signals generated through the subject’s motor imagery (MI). The display screen in the device established a visual feedback platform between the system and the patient. When the patient intended to perform upper/lower limb exercise training, the researcher needed to set the virtual character on the display screen to the corresponding exercise preparation. Thus, in upper limb training, the virtual character was displayed for swimming preparation, whereas for lower limb training, the display was for cycling preparation. The patient carried out the corresponding motor imagery tasks according to the virtual character training type on the display screen and followed the voice prompt. When the patient’s Mscore [35] (the specific EEG signals) reached a certain threshold, the virtual character on the display screen began to move accordingly. At the same time, the rotary shafts of the system began to rotate to drive the patient’s limbs to move accordingly. When the patient’s motor intention fell below the set threshold, the virtual character on the display screen stopped its corresponding motion, and the rotary shafts stopped rotating, thus ceasing to drive the patient’s limbs.

The system gave patients real-time visual and motor perception feedback, that is, during the training, the progress bar on the right side of the display screen showed the intensity of the patient’s motor imagery in real time (updates every 110 ms); when the intensity of the patient’s motor imagery (i.e., Mscore) reached a certain threshold, the virtual character on the display screen would start corresponding movements (swimming or cycling), and the virtual character’s movement would generate visual feedback; meanwhile, the rotary shafts of the system drove the patient’s limb movement (update rate > 1 Hz), thus giving the patient motor perception feedback [35,36].

### 2.3. Training and Evaluation Methods

#### 2.3.1. Feasibility and Safety Verification of the System

First, three stroke patients with hemiplegia were recruited to perform L-B300 system rehabilitation training on a routine basis to verify the feasibility and safety of this system. The training method: with upper and lower limb training twice per day (15 min of upper limb training and 15 min of lower limb training per session, each in the morning and afternoon); thus, a total of 30 min for upper limb training and 30 min for lower limb training per day. The three patients thus completed four training session during two consecutive days. The positioning of a patient during upper and lower limb BCI training is shown in Figure 2.

We evaluated the feasibility of the system by first recording the continuous motor state switching time (CMSST) of each patient when using BCI; this refers to the time taken by the patient to switch from a resting state to a concentrated motor intention state that can continuously drive the system [35]. We also recorded the motor state percentage (MSP), which refers to the proportion of time when the patient’s motor intention exceeds the selected threshold [35], and the effective training revolution (ETR), which is the number of times that the patient commands the robot to rotate during each training session [35].

#### 2.3.2. Effectiveness Evaluation of the System

Upon finding in the validation study in the three stroke patients with hemiplegia that the L-B300 system was feasible and safe to use for rehabilitation training, we proceeded to study the rehabilitation training of a separate group of five stroke patients with hemiplegia. The rehabilitation training method for these five patients was much as in the feasibility and safety study, but they trained for six days a week to a total of complete 12–14 days; thus, 24–28 sessions of training. We measured limb motor function and brain function indices of these patients at baseline and after rehabilitation treatment.

Detection of limb motor function: (1) to determine sitting balance ability, we used the three-level method to evaluate static sitting balance (level 1), self-dynamic sitting balance (level 2), and other-dynamic sitting balance (level 3). Here, static sitting balance refers to the process of maintaining the stability of sitting posture without the influence of external forces when the participant opens his/her eyes. Self-dynamic sitting balance refers to the process by which the patient can adjust from one posture to another and maintain balance without the influence of external forces. Other-dynamic sitting balance refers to the process whereby the body can quickly adjust its center of mass and posture to maintain balance when the center of gravity of the body changes under an external force [37]. (2) The Fugel–Meyer Assessment (FMA) was used to evaluate the motor function of the upper and lower limbs, where the maximum total score is 100; higher scores indicate better limb motor function [38]. (3) The 10 m walking speed test (m/s) was applied only in those patients who could walk independently. The subjects were asked to walk along a 10 m straight line on a level ground at their fastest speed, which was recorded [39]. (4) A 6 min walking distance (6MWD) is the most widely used clinical submaximal exercise test to evaluate systematic and complete responses during exercise. During the test, subjects were asked to walk back and forth at their fastest speed for six minutes along a 30 m straight and level course, and their total walking distance was recorded [40]. (5) The Modified Barthel Index (MBI) was used to assess their ability to complete activities of daily living, with a total score of 100; higher the score indicated better ability to complete activities of daily living [41]. (6) Surface electromyography (sEMG) signal was acquired by using the Delsys^®^ Trigno wireless EMG acquisition system (Delsys Inc., Natick, MA, USA) and EMGWorks^®^ Acquisition software (version 4.7.6, Delsys Inc., Natick, MA, USA). Test parameters: common mode rejection ratio (CMRR) > 80 dB, noise < 750 nV RMS, analog/digital conversion was 16 bit, sampling frequency was 2000 Hz, each data collection time was 5s, used band-pass filter in the analysis software, bandwidth 20–450 Hz, passband ripple <2%, effective measurement range was ±8000 μV, available channel number was 8. Recorded the sEMG signals of the following muscles: biceps brachialis, triceps brachialis, flexor digitorum, extensor digitorum, abductor pollicis brevis, quadriceps femoris, hamstring muscle, anterior tibial muscle, and triceps surae on the hemiplegic side during maximum isometric contraction (MIVC). From these recordings, we selected the root mean square (RMS) values of the 1s peaks [42]. The values of the 1s peaks referred to the maximum values of the 1s regions containing the strongest signals in the 5s sEMG signals collected. We recorded the sEMG signals sequentially in triplicate, and then calculated the average values. 

In the brain function test, the motor evoked potential (MEP) of the M1 region (primary motor cortex) in the cerebral hemisphere both on the lesion side and non-lesion side was collected using a transcranial magnetic stimulator (CR Technology Co., Ltd., Daejeon, Korea) with an 8-wire coil (12.5 cm in diameter, 3.0 T in maximum intensity) [43]. The latent period and amplitude were calculated to detect the conduction of the efferent nerve pathway and the excitability of the underlying cerebral cortex.

### 2.4. Statistical Analysis

Data were analyzed using the MATLAB 2021a. Quantitative variables were expressed as mean ± standard deviation (mean ± SD). The feasibility and safety evaluation data before and after intervention were statistically compared in terms of numerical size. Cohen’s w was used to report the effect size, and thresholds for small, medium, and large effects were 0.10, 0.30, and 0.50, respectively [44].

## 3. Results

### 3.1. Feasibility and Safety Verification of the System

All three testing phase patients received four training sessions of the system over two days. The first two sessions and the second two sessions for each patient were counted as stage 1 and stage 2, respectively. In the four training sessions, the CMSST of the three patients was 17.8 ± 21.0 s (range 0.1 to 50.7 s), and the indicator value at stage 2 was lower than that in stage 1, as shown in Figure 3.

The analysis results of the MSPs and the ETRs per minute of the initial three patients in upper and lower limb training are in Figure 4 and Figure 5. The MSPs in upper and lower limb training were 52.6 ± 25.7% (maximum 86%, minimum 11%) and 72.4 ± 24.0% (maximum 100%, minimum 39.0%), respectively. The ETRs per minute in upper and lower limb training were 25.8 ± 13.0 (maximum 36.3, minimum 3.2) and 24.8 ± 6.4 (maximum 33.3, minimum 18.6), respectively. Comparing stage 2 with stage 1, there were slight declines of these two indicators in the upper limb training of patients P1 and P3. At stage 2, there was a conspicuous improvement in lower limb training of all three patients compared with stage 1.

During the entire training process, the three pilot patients had no discomfort, recurrence of seizure, or any other adverse events, showing the system to be safe and reliable.

### 3.2. Rehabilitation Effects of the System

#### 3.2.1. Clinical Indicators in the Limb Motor Function of Patients

We calculated the changes in motor function relative to the baseline condition in the five patients with rehabilitation training. The results indicated conspicuous improvements in the patients’ sitting balance ability, upper limb FMA, lower limb FMA, 10 m walking speed, 6MWD, and MBI after treatment. All of the improvements were of large effect size, other than 10 m walking speed (with a small effect size), as shown in Table 2.

#### 3.2.2. The RMSs of the Limb Muscles on the Hemiplegic Side

The RMS changes showed that, compared with the baseline, the RMSs were numerically increased for the biceps brachii, triceps brachii, extensor digitorum, quadriceps femoris, hamstring muscle, anterior tibial muscle and triceps surae of the five stroke patients. However, the mean RMSs of flexor digitorum and abductor pollicis brevis decreased slightly, as shown in Table 3.

#### 3.2.3. Brain Function Test Results of Subjects

The MEP differentials before and after treatment were measured. It was found that the MEP latent periods on the lesion side and non-lesion side were shortened and the amplitudes were enhanced after treatment, All Cohen’s w > 0.4, with more than medium effect size, as shown in Table 4.

## 4. Discussion

In this study, we undertook preliminary clinical experiments to investigate the feasibility, safety, and potential rehabilitation efficacy for restoring limb and brain functions of hemiplegic stroke patients using the new BCI system. This system differs from the traditional active/passive rehabilitation training mode with respect to the multichannel feedback, and adopts the active rehabilitation training mode of bidirectional synchronous stimulation of the central and peripheral nervous systems [13,14]. Using this system requires patients to actively participate in the whole process and focus on their training done through motor imagery. Mscore is the specific EEG signals used in this system to timely evaluate the degree of active target task imagination of patients [45,46]. The Mscore was collected through the brain cap, decoded and transmitted to the terminal devices (the rotary shafts and the display screen) via Bluetooth to control the rotary shafts and the virtual character on the display screen. When Mscore meets with certain requirements, the rotary shafts can drive the patient’s limb movement and provide motor perception feedback to the patient’s body sensation, while simultaneously giving visual feedback to the patient on the display screen [36]. In this study, we first validated the feasibility and safety of the novel system in a pilot test of three stroke patients with hemiplegia. We then proceeded to test the rehabilitative efficacy of the system with 24–28 training sessions in another group of five hemiplegic stroke patients.

In the feasibility verification, we used three indicators (CMSST, MSP, and ETR) to evaluate the usability of the system. Among these indicators, CMSST represents the speed whereby patients enter the active training state. In the pilot study, the mean value of the CMSST was 17.8 s, and this indicator was shortened at stage 2 compared to stage 1, indicating that all three patients could quickly adapt to the rehabilitation training of the system [47]. The factor affecting the MSP is mainly the degree of active participation of the patient. In this study, the mean MSP values of the upper and lower limbs of patients were 52.6 and 72.4%, respectively, which revealed that the three stroke patients with hemiplegia could maintain continuous and high-intensity active rehabilitation training instead of giving up training because of fatigue [47]. The ETRs per minute of the system reflects the amount of exercise of a patient [47]. In this study, the average ETRs per minute in the upper and lower limbs of the patients were 25.8 and 24.8, respectively, suggesting that the upper and lower limbs of the patients were fully trained. As observed in Figure 3 and Figure 4, CMSST, MSPs, and ETRs of the three patients differed greatly, and in MSPs and ETRs, stage 2 of the first and third patients was less than stage 1, while stage 2 of the second patient was larger than stage 1. As can be seen from Figure 5, in MSPs and ETRs, there were not many differences among the three patients, and stage 2 of the three patients was all greater than stage 1. These above results were considered to be related to the fact that cycling is relatively easier to imagine than swimming (since most Chinese people have cycling experience, while many do not have swimming experience) [48]. Therefore, there was no great difference in the performance of the three patients on the task of imagining lower limb cycling, while the imagining upper limb swimming task not only showed greatly different performances among the three patients, but also showed unstable performances in different stages of the same patient [47]. In conclusion, all three patients in the pilot study could effectively complete the training task. More importantly, no discomfort, seizure, or other adverse events occurred in association with the study. Thus, stroke patients with hemiplegia could use the system safely and reliably.

Regarding the efficacy of the system for rehabilitation training, we evaluated motor function at baseline and after 12–14 days of rehabilitation treatment in a group of five patients. Scores numerically improved relative to pretreatment baseline, with respect to sitting balance ability, upper limb FMA, lower limb FMA, 10 m walking speed, 6MWD, and MBI, reflecting an improvement in limb motor function and daily living abilities of patients. The sEMG results showed that the RMSs of all these tested muscles except for flexor digitorum and abductor pollicis brevis rose after treatment, indicating that the intervention boosted the vitality of these muscle groups. The results of MEP differences before and after training displayed that the MEP latent periods on the lesion side and non-lesion side were shortened and the amplitudes were increased, signifying that the functions of bilateral cerebral hemispheres were strengthened after treatment [49,50]. The short-term training using the novel BCI system adopted by these stroke patients with hemiplegia could produce multidimensional effects, which should be related to the system’s ability to enhance bidirectional stimulation. Enhancing bidirectional stimulation is the core of the BCI system used in this study. The BCI system selects the characteristic EEG signals related to motor intention during training as the main control index, which is helpful to guide patients to continuously and actively give downward motor control signals. On the one hand, such a design can accurately apply the downward control signals to the peripheral neuromuscular system; on the other hand, repeated high-intensity training of the peripheral neuromuscular system may also improve the function in certain brain areas through visual and motor feedback, thus achieving “bidirectional stimulation” [51]. According to Hebb’s theory (Hebb, 1949) [52], such bidirectional stimulation can cause neural reorganization including the reorganization of brain region excitability and brain networks in humans, thereby improving cognitive, language, and motor functions.

## 5. Limitations

This paper is a case report of 5 BCI rehabilitation training instead of a randomized controlled study. The sample size is relatively small, and the condition of each patient was different, so this report did not conduct paired sample t-tests of pre- and post-intervention data, and only the Cohen’s w values were shown. In addition, the CMSST, MSP, and ETR values of each patient were tested only once before and after training without repeated measurements, and this may be one of the reasons for the large variation of these indicators in the three patients during the feasibility verification; moreover, even in discussing rehabilitation effects of the system, the 5 patients only received 12–14 days of rehabilitation treatment, thus 24–28 sessions of training, which means short treatment courses. All of these are the limitations of this paper, which need to be improved in future studies.

## 6. Conclusions

In summary, the new BCI system including visual and motor feedback is applicable in a program of active rehabilitation training of stroke patients with hemiplegia. It is not only safe, but also enables the patients to enter the active-training state quickly, and attain a sufficient training intensity. In this pilot study, responses after a short course of treatment show promise for therapeutic effects in multiple dimensions, which must be established in a randomized controlled study of a much larger patient population, with follow-up after active training sessions.

## Figures and Tables

**Figure 1 brainsci-12-01083-f001:**
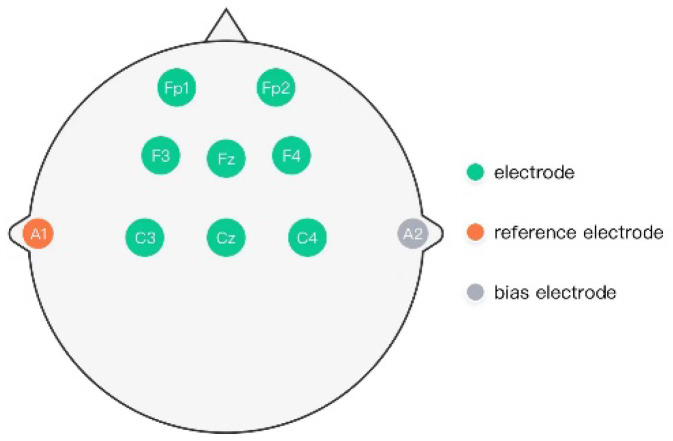
Electrode placement diagram of L-B300 EEG Acquisition and Rehabilitation Training System.

**Figure 2 brainsci-12-01083-f002:**
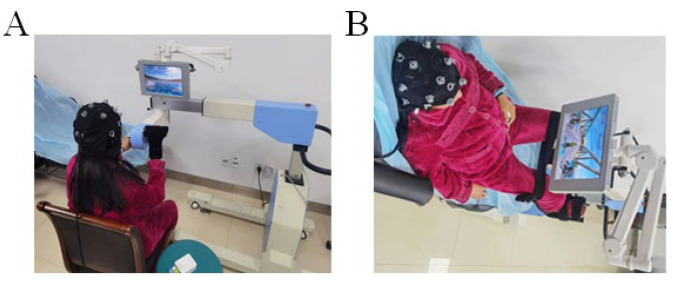
BCI training. (**A**) A patient with right hemiplegia undergoing upper limb BCI training (imagined swimming) (**B**) The same patient undergoing lower limb BCI training (imagined cycling). In the safety evaluation of the system, the three patients were observed and evaluated for any discomfort, seizures, recurrent cerebral hemorrhage, and cerebral infarction during training.

**Figure 3 brainsci-12-01083-f003:**
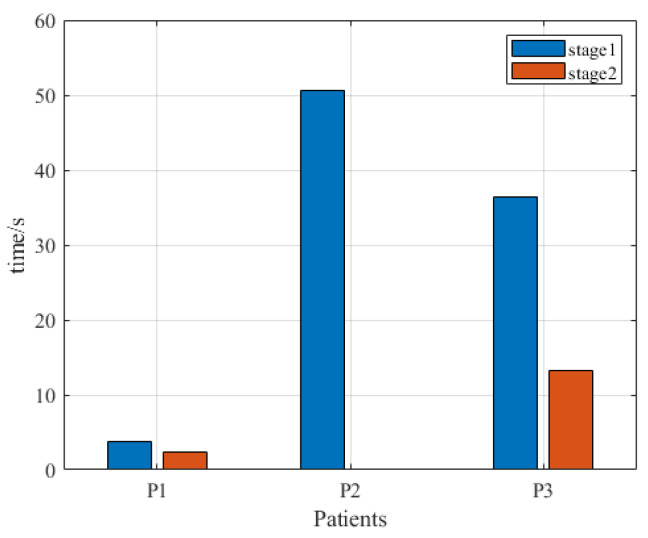
The continuous motor state switching times of the three patients in the feasibility test.

**Figure 4 brainsci-12-01083-f004:**
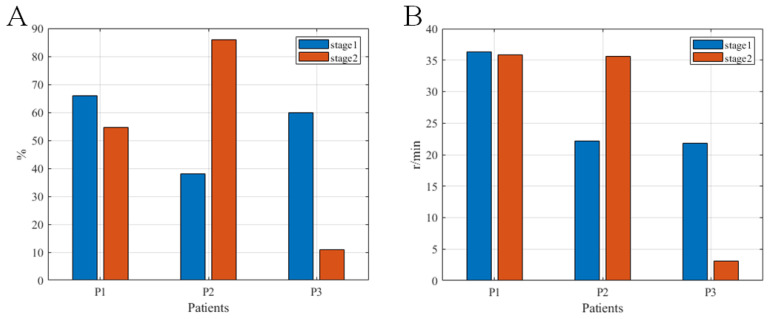
The motor state percentages (**A**) and the effective training revolutions per minute (**B**) of the three test patients in upper limb training at the two stages of training.

**Figure 5 brainsci-12-01083-f005:**
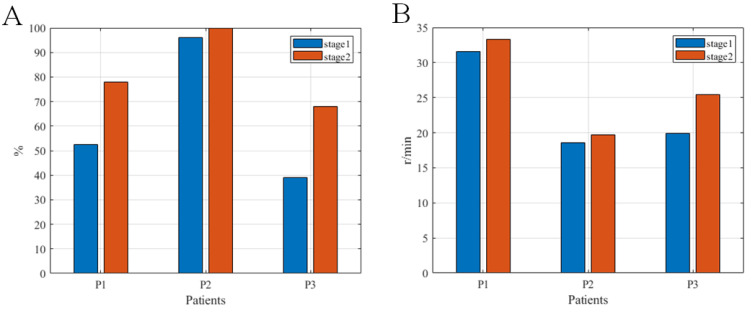
The motor state percentage (**A**) and the effective training revolutions per minute (**B**) of the three pilot patients at the two stages of lower limb training.

**Table 1 brainsci-12-01083-t001:** Information of the eight patients.

Subject	Age	Gender	Sites of Injury	Course of Disease(d)	BCI Treatment Times
1	58	Male	Left basal ganglia cerebral hemorrhage	34	4
2	31	Male	Right basal ganglia cerebral hemorrhage	127	4
3	55	Female	Right basal ganglia cerebral infarction	30	4
1′	34	Male	Left cerebellar hemisphere hemorrhage	41	26
2′	44	Male	Left basal ganglia cerebral hemorrhage	37	26
3′	51	Male	Left basal ganglia cerebral infarction	18	28
4′	52	Female	left basal ganglia cerebral infarction	15	24
5′	58	Male	Left basal ganglia, left paraventricular cerebral infarction	41	26

**Table 2 brainsci-12-01083-t002:** Clinical efficacy evaluation (mean ± SD) of the five patients.

Assessment Item	Before	After	Difference	Cohen’s w
Sitting balance ability	2.4 ± 0.9	2.8 ± 0.5	0.4 ± 0.55	1.22 **
Upper limb FMA	22.2 ± 10.1	30.2 ± 8.9	8.0 ± 5.61	5.18 **
Lower limb FMA	19.6 ± 10.1	25.0 ± 5.7	5.4 ± 5.18	5.44 **
10 m walking speed	0.5 ± 0.2	0.5 ± 0.2	0.04 ± 0.03	0.16
6MWD	155.6 ± 50.0	167.0 ± 48.9	11.4 ± 4.30	2.29 **
MBI	65.0 ± 7.1	72.0 ± 7.6	7.0 ± 2.70	2.06 **

** (Cohen’s w ≥ 0.5) indicates a large effect size.

**Table 3 brainsci-12-01083-t003:** Comparison of the RMSs (mean ± SD) before and after treatment in the five patients.

Assessment Item	Examined Position	Before	After	Difference	Cohen’s w
RMS (μV)	Biceps brachii	2.9 ± 1.1	3.0 ± 1.1	0.2 ± 0.09	0.25
	Triceps brachii	6.8 ± 7.7	7.0 ± 7.9	0.2 ± 0.25	0.20
	Flexor digitorum	0.6 ± 0.5	0.6 ± 0.4	0.0 ± 0.07	0.11
	Extensor digitorum	0.7 ± 0.8	0.7 ± 0.8	0.0 ± 0.01	0.06
	Abductor pollicis brevis	0.3 ± 0.2	0.3 ± 0.2	0.0 ± 0.03	0.08
	Quadriceps femoris	18.5 ± 12.5	19.0 ± 12.8	0.5 ± 0.68	0.35 *
	Hamstring muscle	15.1 ± 8.0	15.5 ± 8.2	0.4 ± 0.35	0.24
	Anterior tibial muscle	9.0 ± 6.4	9.1 ± 6.5	0.2 ± 0.14	0.14
	Triceps surae	10.0 ± 6.5	10.8 ± 7.6	0.9 ± 1.06	0.71 **

** (Cohen’s w ≥ 0.5) indicates a large effect size, * (Cohen’s w < 0.5 and ≥0.3) indicates a medium effect size.

**Table 4 brainsci-12-01083-t004:** Comparison of the MEPs (mean ± SD) before and after treatment in the five patients.

Assessment Item	Examined Position	Testing Indicator	Before	After	Difference	Cohen’s w
MEP	M1 area on the non-lesion side	Latent period (ms)	42.1 ± 8.3	39.5 ± 8.6	−2.6 ± 2.17	1.11 **
		Amplitude (10^−5^)	32.7 ± 10.9	36.2 ± 9.0	3.6 ± 3.78	2.41 **
	M1 area on the lesion side	Latent period (ms)	14.4 ± 5.7	13.8 ± 5.9	−0.6 ± 0.39	0.44 *
		Amplitude (10^−5^)	20.0 ± 7.7	23.9 ± 7.8	3.7 ± 3.89	2.83 **

** (Cohen’s w ≥ 0.5) indicates a large effect size, * (Cohen’s w < 0.5 and ≥0.3) indicates a medium effect size.

## Data Availability

Data are available upon request.

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
