# Peer review of "Application of a Brain–Computer Interface System with Visual and Motor Feedback in Limb and Brain Functional Rehabilitation after Stroke: Case Report"

_brainsci, 2022, doi:10.3390/brainsci12081083_

Round 1
Reviewer 1 Report
I have no comments for this paper. This is well written, clear design, results could be presented better than just the bar graphs, i.e., probably a single bar graph combining all the metrics. Please mention whether patients were consented for this study.
Author Response
(1)results could be presented better than just the bar graphs, i.e., probably a single bar graph combining all the metrics.
Response:Thank you very much for your valuable advice.
Stage 1 and Stage 2 are described separately, which can better show the changes of each index with the increase of training
times; in addition, separate description of the upper and lower limbs can more clearly show the difference of each index
between the upper and lower limbs training through the BCI system. In view of this, can we ask the expert to approve our
present expression ? thank you very much.
(2)Please mention whether patients were consented for this study.
Response:Thank you very much for your valuable advice.
This sentence of “Written informed consent was obtained from each participant before enrollment.” has been added in
the last paragraph of “2.1 Subjects”.
Reviewer 2 Report
Your work seems to me of great merit.
It is true that the number of the sample is very small and has a series of limitations that must be included in the paper for its publication.
Author Response
Response:Thank you very much for your valuable advice.
The limitations of the current experiments design and results have been described in “5. Limitations”
Reviewer 3 Report
Dear authors
Thank you for your valuable research.
There are important to mention some points about increasing the quality level of the article.
Please:
1- In the title of the article, replace "A Pilot study" with "Case Report".
2- In the introduction section, mention the complications caused by stroke and lengthening the rehabilitation time and the role of rehabilitation through BCI in stroke or similar disorders (preferably upper motor neuron diseases).
3- In the discussion of the study, only a few limited references have been used. For example, for the first two paragraphs of the discussion, which are long, only one reference is used. You are expected to support your results by citing similar studies using BCI. Also, strengthen the discussion by using the basic neural mechanisms and referring to the neural networks involved in the mentioned patients
Author Response
1- In the title of the article, replace "A Pilot study" with "Case Report".
Response:Thank you very much for your valuable advice.
In the title of the article, "Case Report" has been used.
2- In the introduction section, mention the complications caused by stroke and lengthening the
rehabilitation time and the role of rehabilitation through BCI in stroke or similar disorders (preferably
upper motor neuron diseases).
Response:Thank you very much for your valuable advice.
Some relevant content has been added to the third paragraph of “1. Introduction”.
3- In the discussion of the study, only a few limited references have been used. For example, for the
first two paragraphs of the discussion, which are long, only one reference is used. You are expected
to support your results by citing similar studies using BCI. Also, strengthen the discussion by using
the basic neural mechanisms and referring to the neural networks involved in the mentioned
patients
Response:Thank you very much for your valuable advice.
Some relevant content has been added and the possible neural mechanisms have been analysed in “4. Discussion”.
Reviewer 4 Report
In this paper, the authors evaluated a brain-computer interface (BCI) system in upper and lower-limb rehabilitation training on individuals with post-stroke. Three and five stroke patients were recruited in the feasibility testing phase and long-term rehabilitation training phase, respectively. Comparison results of multiple evaluation metrics before training and after training were presented to show the benefits of rehabilitation training by using the BCI system. However, the writing of the manuscript needs to be significantly improved. The motivations and innovations of the paper were not well explained. The results analysis and discussion were not well organized, and more deep interpretation and evaluation are required. More specific comments from the reviewer can be found in the attached file.
The results description in the abstract only covered quantitative analysis from the three patients in the feasibility
and safety verication phase, but what about the quantitative analysis for the give hemiplegic stroke patients in
the rehabilitation phase? The results analysis of the later group should be the focus of this paper.
ˆ The literature review related to interventions for stroke treatment and rehabilitation needs to be improved to cover
more modern technologies, including powered exoskeletons and hybrid neuroprosthesis (FES + powered exoskele-
tons) in those papers such as :
1) Zhang, Q., Sun, D., Qian, W., Xiao, X. and Guo, Z., 2020. Modeling and control of a cable-driven rotary series
elastic actuator for an upper limb rehabilitation robot. Frontiers in neurorobotics, 14, p.13.
2) Awad, L.N., Lewek, M.D., Kesar, T.M., Franz, J.R. and Bowden, M.G., 2020. These legs were made for
propulsion: advancing the diagnosis and treatment of post-stroke propulsion decits. Journal of NeuroEngineering
and Rehabilitation, 17(1), pp.1-16.
3) Takahashi, K.Z., Lewek, M.D. and Sawicki, G.S., 2015. A neuromechanics-based powered ankle exoskeleton to
assist walking post-stroke: a feasibility study. Journal of neuroengineering and rehabilitation, 12(1), pp.1-13.
4) Molazadeh, V., Zhang, Q., Bao, X. and Sharma, N., 2021. An iterative learning controller for a switched
cooperative allocation strategy during sit-to-stand tasks with a hybrid exoskeleton. IEEE Transactions on Control
Systems Technology, 30(3), pp.1021-1036.
ˆ The motivation of this current study was not clear. What are these challenges of the current BCI implementations
related to the treatment and rehabilitation after stroke? What the questions this paper wants to address?
ˆ In the inclusion criteria, what does the MMT mean? How to calculate this score? Please at least add a reference
there.
ˆ In the statement of The electrode placement was in line with the 10/20..., please add a reference here for a better
clarication.
ˆ Please explain why two modes were applied for the rehabilitation training system? How did them relate to the
visual and motor feedback?
ˆ The writing needs to be polished carefully, especially many typos really aect the understanding for readers, like
'There' in line 197.
ˆ How could the mean and standard deviation of the age from these three patients in the feasibility and safety testing
phase be 53±2, given the age numbers in Table 1? In addition, what does that mean of the course in Table 1?
ˆ The section 3.1 General information of patients was supposed to be in the section 2.1 instead of being the results
section.
ˆ More analysis and explanation are needed for results shown in each gure. For example, in Fig.3, why the variation
of CMSST of the three patients is so high? Why the indicator value at stage 2 is lower than stage 1? It seems like
only one test of CMSST value on each patient before and after training session, and this could be the reason that
the variation is so high. Repeated testing CMSST values need to be conducted to reduce the bias induced by the
one measure. The similar issue also applies to Fig. 4 and Fig. 5.
ˆ Table 2 and Table 3 need to be improved to t the dimension of the page margin.
ˆ The processing and analysis of sEMG signals were not clear, like what the sampling frequency was, how many
seconds data collection, what lters were used, what the 1 second peaks meant, and so on. The RMS results
of sEMG in Table 3 did not have unit. If the mean RMS value of exor digitorum and abductor pollicis brevis
decreased, why the Cohen's w values were still positive?
ˆ There was no statistically signicant dierence analysis throughout the paper. Only the Cohen's w values were
shown.
1
ˆ The authors mentioned the implementation of both virtual and motor feedback, but actually no description of how
the feedback was designed or used in the paper, which really confused the reviewer.
ˆ There was no description related to the 'active rehabilitation training mode of bidirectional synchronous stimulation
of the central and peripheral nervous systems' throughout the paper, but this was summarized in the discussion
section. What does this training mode mean?
ˆ The discussion section needs to be rewritten. Basically, the discussion section repeated the experimental protocol
design in section 2 and results in section 3. It did not add any new interpretation, explanation, or limitation of the
current experiments design and results.
ˆ The format of the references section was not consistent, like the journals names with both full and abbreviated
versions, some page numbers were missing, some typos of authors names, and so on. It needs to be carefully checked
and revised.
Author Response
In this paper, the authors evaluated a brain-computer interface (BCI) system in upper and lower-limb rehabilitation
training on individuals with post-stroke. Three and five stroke patients were recruited in the feasibility testing phase and long-term rehabilitation training phase, respectively. Comparison results of multiple evaluation metrics between before
training and after training were presented to show the benefits of the rehabilitation training by using the BCI system.
However, the writing of the manuscript is poor and needs to be significantly improved. The motivations and innovations
of the paper were not well explained. The results analysis and discussion were not well organized, and more deep
interpretation and evaluation are required. More specific comments from the reviewer can be found below.
1. The results description in the abstract only covered quantitative analysis from the three patients in the feasibility and
safety verification phase, but what about the quantitative analysis for the give hemiplegic stroke patients in the rehabilitation
phase? The results analysis of the later group should be the focus of this paper.
Response:Thank you very much for your valuable advice.
The abstract has been revised according to your valuable advice.
2. The literature review related to interventions for stroke treatment and rehabilitation needs to be improved to cover more
modern technologies, including powered exoskeletons and hybrid neuroprosthesis (FES + powered exoskeletons) in those
papers such as :
1) Zhang, Q., Sun, D., Qian, W., Xiao, X. and Guo, Z., 2020. Modeling and control of a cable-driven rotary series elastic
actuator for an upper limb rehabilitation robot. Frontiers in neurorobotics, 14, p.13.
2) Awad, L.N., Lewek, M.D., Kesar, T.M., Franz, J.R. and Bowden, M.G., 2020. These legs were made for propulsion:
advancing the diagnosis and treatment of post-stroke propulsion decits. Journal of NeuroEngineering and Rehabilitation,
17(1), pp.1-16.
3) Takahashi, K.Z., Lewek, M.D. and Sawicki, G.S., 2015. A neuromechanics-based powered ankle exoskeleton to assist
walking post-stroke: a feasibility study. Journal of neuroengineering and rehabilitation, 12(1), pp.1-13.
4) Molazadeh, V., Zhang, Q., Bao, X. and Sharma, N., 2021. An iterative learning controller for a switched cooperative
allocation strategy during sit-to-stand tasks with a hybrid exoskeleton. IEEE Transactions on Control Systems
Technology, 30(3), pp.1021-1036.
Response:Thank you very much for your valuable advice.
Some modern technologies have been added in the second paragraph of “1.Introduction” and some of the above
references have been cited.
3. The motivation of this current study was not clear. What are these challenges of the current BCI implementations
related to the treatment and rehabilitation after stroke? What the questions this paper wants to address? ˆ
Response:Thank you very much for your valuable advice.
(1)The motivation of this current study has been added in the last paragraph of "1. Introduction”, that is,"The purpose
of this study was to observe the feasibility and safety of this novel BCI system for rehabilitation of stroke patients with
hemiplegia, and the effects of a short course of BCI treatment on patients' limb movement and brain function ”.
(2)What are these challenges of the current BCI implementations related to the treatment and rehabilitation after stroke?
Response:Thank you very much for your valuable advice.
How to make stroke patients effectively imagine target tasks? How to observe the performance of imaginary tasks? How
to accurately capture and identify task-related EEG signals? and how to convert EEG signals into views, movements, sounds,
etc., in order to achieve the enhanced efficacy of BCI training with multiple sensory feedback are still challenges.(The
content has been already added in the first paragraph of “4. Discussion”)
(3)What the questions this paper wants to address?
Response:Thank you very much for your valuable advice.
This paper aims to report that a novel BCI system with both visual and motor feedback can be safely and effectively used
in rehabilitation training of stroke patients with hemiplegia. (The relevant content has been in Conclusions of the
Abstract and in “6. Conclusions”.)
4.In the inclusion criteria, what does the MMT mean? How to calculate this score? Please at least add a reference there. ˆ
Response:Thank you very much for your valuable advice.
MMT (manual muscle testing) is a common tool for evaluating muscle strength in clinical practice. “Inclusion
criteria:(3)” has been revised, and a reference has been added here.
5. In the statement of “The electrode placement was in line with the 10/20...”, please add a reference here for a better
clarification. ˆ
Response:Thank you very much for your valuable advice.
A reference has been added here (“2.2 L-B300 EEG Acquisition and Rehabilitation Training System”).
6. Please explain why two modes were applied for the rehabilitation training system? How did them relate to the visual
and motor feedback? ˆ
(1) Please explain why two modes were applied for the rehabilitation training system?
Response:Thank you very much for your valuable advice.
The system used in this study provides two training modes: mode 1, If the patient has adequate muscle strength, mode 1 can be used, in which the equipment is driven through the collected EEG signals in concert with the patient's own muscle strength; mode 2, if the patient has complete limb weakness or very little muscle strength,mode 2 can be used, in which the equipment is driven completely by the patient's EEG signals. Obviously, mode 2 training is more difficult than mode 1
training. The purpose of adopting mode 2 in this study is to increase the difficulty of training, so as to better verify the
feasibility, safety and effect of training with this system for patients with unilateral brain injury. (The relevant explanations
have been added in “2.2 L-B300 EEG Acquisition and Rehabilitation Training System” and the first paragraph of
“2.3.1 Feasibility and Safety Verification of the System”.)
(2) How did them relate to the visual and motor feedback? ˆ
Response:Thank you very much for your valuable advice.
At the same time, the system gives patients real-time visual and motor perception feedback, that is, during the training,
the progress bar on the right side of the display screen shows the intensity of the patient's motor imagery in real time (updates every 110ms); when the intensity of the patient's motor imagery reaches a certain threshold, the virtual character on the
display screen will start corresponding movements (swimming or cycling), and the virtual character's movement will
generate visual feedback; meanwhile, the rotary shafts of the system began to rotate to drive the patient's limbs to move accordingly(Update rate >1Hz), thus giving the patient motor perception feedback [17].(The information has been added in the second paragraph of “4. Discussion” )
7. The writing needs to be polished carefully, especially many typos really affect the understanding for readers, like
“There ” in line 197. ˆ
Response:Thank you very much for your valuable advice.
We are sorry for the trouble caused by our spelling mistake. We have changed "There" into "Three" in the third
paragraph of “2.1 Subjects”. And the writing has be polished carefully again.
8. How could the mean and standard deviation of the age from these three patients in the feasibility and safety testing
phase be 53±2, given the age numbers in Table 1? In addition, what does that mean of the course in Table 1? ˆ
(1)How could the mean and standard deviation of the age from these three patients in the feasibility and safety testing
phase be 53±2, given the age numbers in Table 1?
Response:Thank you very much for your valuable advice.
We have updated the calculation result with a value of 48±15 (years old) in the third paragraph of “2.1 Subjects”.
(2)In addition, what does that mean of the course in Table 1?
Response:Thank you very much for your valuable advice.
Course means course of disease. We have changed Course to “Course of Disease(d)” in Table 1.
9. The section 3.1 General information of patients was supposed to be in the section 2.1 instead of being the results
section. ˆ
Response:Thank you very much for your valuable advice.
We have adjusted the position of this section.
10. More analysis and explanation are needed for results shown in each figure. For example, in Fig.3, why the variation of CMSST of the three patients is so high? Why the indicator value at stage 2 is lower than stage 1? It seems like only one test of CMSST value on each patient before and after training session, and this could be the reason that the variation is so high. Repeated testing CMSST values need to be conducted to reduce the bias induced by the one measure. The similar issue also applies to Fig. 4 and Fig. 5.
ˆ
(1)For example, in Fig.3, why the variation of CMSST of the three patients is so high?
Response:Thank you very much for your valuable advice.
Figures 3-5 show the CMSSTs, MSPs and ETRs of the three patients were obviously different, which are thought to be
mainly related to the states of the patients at that time; in addition, CMSST, MSP and ETR values were tested only once
for each patient before and after training, without repeated measurements, which may also be one reason for the large
variation. ( These reasons are analyzed in the third Paragraph in "4. Discussion")
(2)Why the indicator value at stage 2 is lower than stage 1?
Response:Thank you very much for your valuable advice.
There were slight declines of the MSPs and ETRs in the upper limb training of patients P1 and P3 at stage 2 compared to
those at stage 1, which might reflect the training states of these two patients at that time. (The possible causes has been
analyzed in the third Paragraph in "4. Discussion")
11. Table 2 and Table 3 need to be improved to fit the dimension of the page margin. ˆ
Response:Thank you very much for your valuable advice.
The positions of Tables 2 and 3 have been adjusted.
12. The processing and analysis of sEMG signals were not clear, like what the sampling frequency was, how many
seconds data collection, what filters were used, what the 1 second peaks meant, and so on. The RMS results of sEMG in
Table 3 did not have unit. If the mean RMS value of flexor digitorum and abductor pollicis brevis decreased, why the
Cohen ”s w values were still positive? ˆ
Response:Thank you very much for your valuable advice.
(1)Relevant information about “The processing and analysis of sEMG signals” has been added in the second Paragraph
of “2.3.2 Effectiveness Evaluation of The System”.
(2) The units of the RMS results in Table 3 have been already labeled as μV.
(3)If the mean RMS value of flexor digitorum and abductor pollicis brevis decreased, why the Cohen ”s w values were still
positive? ˆ
Response:Thank you very much for your valuable advice.
In our calculations, the Cohen's w is a non-negative value, as detailed in the reference[31].
13. There was no statistically significant difference analysis throughout the paper. Only the Cohen ”s w values were
shown.
Response:Thank you very much for your valuable advice.
Because the sample of this report is small and the indicators between the patients are quite different, in this case,
Cohen's W is more appropriate to compare the data. In this report, the mean values have been compared and the effects
have been calculated. The comprehensive results show that the sitting balance ability, upper limb Fugel-Meyer assessment
(FMA), lower limb FMA, 6 minute walking distance, modified Barthel index, RMS of triceps surae, and three of the MEP
parameters were improved after treatment, and all these indicators had a large effect size (Cohen’s w ≥ 0.5 ), showing
potential multidimensional benefits after a short course of BCI training.
14. The authors mentioned the implementation of both virtual and motor feedback, but actually no description of how the
feedback was designed or used in the paper, which really confused the reviewer.
Response:Thank you very much for your valuable advice.
At the same time, the system gives patients real-time visual and motor perception feedback, that is, during the training,
the progress bar on the right side of the display screen shows the intensity of the patient's motor imagery in real time
(updates every 110ms); when the intensity of the patient's motor imagery reaches a certain threshold, the virtual character
on the display screen will start corresponding movements (swimming or cycling), and the virtual character's movement
will generate visual feedback; meanwhile, the rotary shafts of the system began to rotate to drive the patient's limbs to
move accordingly(update rate >1Hz), thus giving the patient motor perception feedback.(The content has been added
in the second paragraph of "4. Discussion")
15. There was no description related to the “active rehabilitation training mode of bidirectional synchronous stimulation of
the central and peripheral nervous systems ” throughout the paper, but this was summarized in the discussion section.
What does this training mode mean? ˆ
Response:Thank you very much for your valuable advice.
The information about the bidirectional stimulation provided by this system has been added to the last paragraph in
“4. Discussion”.
16. The discussion section needs to be rewritten. Basically, the discussion section repeated the experimental protocol
design in section 2 and results in section 3. It did not add any new interpretation, explanation, or limitation of the current
experiments design and results.
Response:Thank you very much for your valuable advice.
(1)Much of the discussion section has been revised, and a lot of content has been added to the discussion section.
(2)The limitations of the current experiments design and results have been described in “5. Limitations”
17. The format of the references section was not consistent, like the journals names with both full and abbreviated
versions, some page numbers were missing, some typos of authors names, and so on. It needs to be carefully checked and
revised.
Response:Thank you very much for your valuable advice.
The format of the references section has been revised carefully and is consistent.
Round 2
Reviewer 4 Report
The authors have improved the quality of the manuscript and addressed some concerns from the reviewer. However, some comments have not been addressed correctly and some small issues still exist after checking these changes, which can be found in the attached file.
The authors have added only some of the suggested studies, however, the improved literature review is still not
strong enough to support the gap between the existing studies and the work presented in the current study. For
example, more introduction related to sole visual feedback or sole motor feedback in the BCI system in existing
studies should be reported, since the BCI system with both visual and motor feedback was proposed as a new
approach in the current study. The originally suggested modern technologies for stroke treatment and rehabilitation
include powered exoskeletons and hybrid neuroprosthesis (FES + powered exoskeletons) in those papers like:
1) Zhang, Q., Sun, D., Qian, W., Xiao, X. and Guo, Z., 2020. Modeling and control of a cable-driven rotary series
elastic actuator for an upper limb rehabilitation robot. Frontiers in neurorobotics, 14, p.13.
2) Awad, L.N., Lewek, M.D., Kesar, T.M., Franz, J.R. and Bowden, M.G., 2020. These legs were made for
propulsion: advancing the diagnosis and treatment of post-stroke propulsion decits. Journal of NeuroEngineering
and Rehabilitation, 17(1), pp.1-16.
3) Takahashi, K.Z., Lewek, M.D. and Sawicki, G.S., 2015. A neuromechanics-based powered ankle exoskeleton to
assist walking post-stroke: a feasibility study. Journal of neuroengineering and rehabilitation, 12(1), pp.1-13.
4) Molazadeh, V., Zhang, Q., Bao, X. and Sharma, N., 2021. An iterative learning controller for a switched
cooperative allocation strategy during sit-to-stand tasks with a hybrid exoskeleton. IEEE Transactions on Control
Systems Technology, 30(3), pp.1021-1036.
ˆ It would be better to include these challenges of the current BCI implementations related to the treatment and
rehabilitation after stroke in the introduction section rather than in the discussion section. Again, more current
studies of the state-of-the-art in BCI system with single signal feedback need to be included in the introduction
section.
ˆ It looks like the text font throughout the paper is not consistent, for example, the reference [22] in like 123 when
compared to other reference numbers. Please recheck the format and make all consistent.
ˆ The authors only explained the two modes that were applied for the rehabilitation training system in the response
letter but not in the revised manuscript. Also, what is the purpose of introducing these two modes while the
authors only used the second mode in this study? Again, the authors did not clarify what is the corresponding the
explanation related to the visual and motor feedback needs to be presented in the Materials and methods section
rather than in the discussion section.
ˆ In addition, does the (d) mean number of day in Table 1? If so, according to the literature, people with chronic
stroke > 6 months were usually included in existing studies, then what would be the reason for including short
period course of disease in the current study?
ˆ More analysis and explanation are needed for results shown in each gure. For example, in Fig.3, why the variation
of CMSST of the three patients is so high? Why the indicator value at stage 2 is lower than stage 1? It seems like
only one test of CMSST value on each patient before and after training session, and this could be the reason that
the variation is so high. Repeated testing CMSST values need to be conducted to reduce the bias induced by the
one measure. The similar issue also applies to Fig. 4 and Fig. 5. Basically, this comment was not addressed in this
revision, and nothing new is added in the revised results section. The specic result discussion in each gure needs
to be given point-by-point for more clear explanations rather than putting all together within one sentence in the
discussion section as what the authors did in this revision.
ˆ The processing and analysis of sEMG signals were not clear, like what the sampling frequency was, how many
seconds data collection, what lters were used, what the 1 second peaks meant, and so on. The parameters of the
band-pass lter were not provided in the revised manuscript.
ˆ There was no statistically signicant dierence analysis throughout the paper. Only the Cohen's w values were
shown. What would be the reason to select this small sample size in the current study? Have the authors done any
power analysis to determine the minimal sample size?
ˆ Although the authors explained the implementation of both virtual and motor feedback in the discussion section,
the signals to generate the motor imagery or visual feedback were not clearly explained. Also, these clarications
need to be included in the Materials and methods section.
ˆ Did the BCI system in the current study include FES during the rehabilitation training procedure? If yes, please
provide more details of the implementation of the FES system; if not, please clearly specify the dierence between
the BCI system in the current study and other BCI system with FES.
Author Response
The authors have improved the quality of the manuscript and addressed some
concerns from the reviewer.
1. The authors have added only some of the suggested studies, however, the
improved literature review is still not strong enough to support the gap
between the existing studies and the work presented in the current study.
For example, more introduction related to sole visual feedback or sole
motor feedback in the BCI system in existing studies should be reported,
since the BCI system with both visual and motor feedback was proposed
as a new approach in the current study. The originally suggested modern
technologies for stroke treatment and rehabilitation include powered
exoskeletons and hybrid neuroprosthesis (FES + powered exoskeletons) in
those papers like:
1) Zhang, Q., Sun, D., Qian, W., Xiao, X. and Guo, Z., 2020. Modeling and control of a cable-driven rotary series elastic actuator for an upper limb
rehabilitation robot. Frontiers in neurorobotics, 14, p.13.
2) Awad, L.N., Lewek, M.D., Kesar, T.M., Franz, J.R. and Bowden, M.G., 2020. These legs were made for propulsion: advancing the diagnosis and treatment
of post-stroke propulsion decits. Journal of NeuroEngineering and Rehabilitation, 17(1), pp.1-16.
3) Takahashi, K.Z., Lewek, M.D. and Sawicki, G.S., 2015. A neuromechanics-based powered ankle exoskeleton to assist walking post-stroke: a feasibility
study. Journal of neuroengineering and rehabilitation, 12(1), pp.1-13.
4) Molazadeh, V., Zhang, Q., Bao, X. and Sharma, N., 2021. An iterative learning controller for a switched cooperative allocation strategy during sit-to-stand
tasks with a hybrid exoskeleton. IEEE Transactions on Control Systems Technology, 30(3), pp.1021-1036. ˆ
Response:Thank you very much for your valuable advice.
Most BCI rehabilitation systems adopt single feedback modality, such as systems based on vision [18-20], kinesthetic
sense [21-23], or perception[24,25]. The main purpose is to provide perceptual or sensory feedback, and these feedback
modalities can lead to motor function recovery. How these feedback patterns affect clinical outcomes remains elusive.
However, a study has shown that sensory feedback of exercise may be a key factor in BCI-based rehabilitation, and single
visual feedback is not sufficient to arouse functional gain [26]. In addition, immersive visual feedback may help enhance
the effect of feedback [27,28]. (The fourth paragraph in “1. Introduction”)
Compared with other BCI rehabilitation systems, the BCI rehabilitation system used in this report has the advantage
of both visual and motor feedback and high time efficiency: its real-time feedback is only 110 ms, and this parameter is
9
smaller than those of other systems (300 ms [29] and 200 ms [30] ) ; it is so responsive and user-friendly that the user is not
easy to feel tired [31]; it is very convenient in clinical use and can guide and timely monitor target task imagination of
patients. The purpose of this study was to observe the feasibility and safety of this novel BCI system for rehabilitation of
stroke patients with hemiplegia, and the effects of a short course of BCI treatment on patients' limb movement and brain
function. (The last paragraph in “1. Introduction”)
2. It would be better to include these challenges of the current BCI implementations related to the
treatment and rehabilitation after stroke in the introduction section rather than in the discussion
section. Again, more current studies of the state-of-the-art in BCI system with single signal feedback
need to be included in the introduction section. ˆ
Response:Thank you very much for your valuable advice.
These challenges of the current BCI implementations related to the treatment and rehabilitation after stroke have been
adjusted to the introduction section. (the third paragraph of “1. Introduction”)
Most BCI rehabilitation systems adopt single feedback modality, such as systems based on vision [18-20], kinesthetic
sense [21-23], or perception[24,25]. The main purpose is to provide perceptual or sensory feedback, and these feedback
modalities can lead to motor function recovery. How these feedback patterns affect clinical outcomes remains elusive.
However, a study has shown that sensory feedback of exercise may be a key factor in BCI-based rehabilitation, and single
visual feedback is not sufficient to arouse functional gain [26]. In addition, immersive visual feedback may help enhance
the effect of feedback [27,28]. (The fourth paragraph in “1. Introduction”)
Compared with other BCI rehabilitation systems, the BCI rehabilitation system used in this report has the advantage
of both visual and motor feedback and high time efficiency: its real-time feedback is only 110 ms, and this parameter is
smaller than those of other systems (300 ms [29] and 200 ms [30] ) ; it is so responsive and user-friendly that the user is not
easy to feel tired [31]; it is very convenient in clinical use and can guide and timely monitor target task imagination of
patients. The purpose of this study was to observe the feasibility and safety of this novel BCI system for rehabilitation of
stroke patients with hemiplegia, and the effects of a short course of BCI treatment on patients' limb movement and brain
function. (The last paragraph in “1. Introduction”)
2. It looks like the text font throughout the paper is not consistent, for example, the reference [22]
in like 123 when compared to other reference numbers. Please recheck the format and make all
consistent.
ˆ
Response:Thank you very much for your valuable advice.
The format has been rechecked and adjusted carefully.
3. The authors only explained the two modes that were applied for the rehabilitation training
system in the response letter but not in the revised manuscript. Also, what is the purpose of
introducing these two modes while the authors only used the second mode in this study? Again,
the authors did not clarify what is the corresponding the explanation related to the visual and
10
motor feedback needs to be presented in the Materials and methods section rather than in the
discussion section. ˆ
Response:Thank you very much for your valuable advice.
Since mode 2 was not used in this study, it has been deleted from this paper. (in the first paragraph of “2.2 L-B300
EEG Acquisition and Rehabilitation Training System”)
The explanation related to the visual and motor feedback has be presented in the second paragraph of “2.2 L-B300
EEG Acquisition and Rehabilitation Training System”.
4. In addition, does the (d) mean number of day in Table 1? If so, according to the literature, people
with chronic stroke > 6 months were usually included in existing studies, then what would be
the reason for including short period course of disease in the current study? ˆ
Response:Thank you very much for your valuable advice.
Indeed, (d) in Table 1 means number of day.
This is because our country is promoting the implementation of the "tertiary rehabilitation network" plan, that is, the
national tertiary hospitals mainly undertake the rehabilitation of difficult, urgent and severe diseases, while the rehabilitation
of common and frequently occurring chronic diseases is mainly undertaken by secondary and below hospitals or
communities. Our hospital is a national level three grade A hospital, so there are more stroke patients in the acute phase and
early recovery, and this is the reason for the short overall disease duration of the subjects in our study. In future studies, we
will pay attention to the inclusion of stroke patients with longer disease duration through multi-center cooperation.
5. More analysis and explanation are needed for results shown in each figure. For example, in Fig.3,
why the variation of CMSST of the three patients is so high? Why the indicator value at stage 2
is lower than stage 1? It seems like only one test of CMSST value on each patient before and after
training session, and this could be the reason that the variation is so high. Repeated testing
CMSST values need to be conducted to reduce the bias induced by the one measure. The similar
issue also applies to Fig. 4 and Fig. 5. Basically, this comment was not addressed in this revision,
and nothing new is added in the revised results section. The specific result discussion in each
figure needs to be given point-by-point for more clear explanations rather than putting all
together within one sentence in the discussion section as what the authors did in this revision. ˆ
Response:Thank you very much for your valuable advice.
All three testing phase patients received four training sessions of the system over two days. The first two sessions and
the second two sessions for each patient were counted as stage 1 and stage 2, respectively. (the first paragraph in “3.1
Feasibility and safety verification of the system”)
Therefore, CMSST, MSPs and the ETRs were the averages of the first two sessions and the second two sessions.
11
As observed in Figure 3 and 4, CMSST, MSPs and ETRs of the three patients differed greatly, and in MSPs and ETRs,
stage 2 of the first and third patients was less than stage 1, while stage 2 of the second patient was larger than stage 1. As
can be seen from Figure 5, in MSPs and ETRs, there were not much differences among the three patients, and stage 2 of the
three patients was all greater than stage 1. These results were considered to be related to the fact that cycling is relatively
easier to imagine than swimming (since most Chinese people have cycling experience, while many have not swimming
experience) [34]. Therefore, there was no great difference in the performance of the three patients on the task of imagining
lower limb cycling, while the imagining upper limb swimming task not only showed greatly different performances among
the three patients, but also showed unstable performances in different stages of the same patient [35]. (The content has
been added in the second paragraph of “4. Discussion”)
6. The processing and analysis of sEMG signals were not clear, like what the sampling frequency
was, how many seconds data collection, what filters were used, what the 1 second peaks meant,
and so on. The parameters of the band-pass filter were not provided in the revised manuscript.
ˆ
Response:Thank you very much for your valuable advice.
Test parameters: common mode rejection ratio (CMRR) > 80 dB, noise < 750 nV RMS, analog / digital conversion was
16 bit, sampling frequency was 2000 Hz, each data collection time was 5 s, used band-pass filter in the analysis software,
bandwidth 20-450Hz, passband ripple <2%, effective measurement range was ± 8 000 μV, available channel number was 8.
(The second paragraph of “2.3.2 Effectiveness evaluation of the system”)
The values of the 1 second peaks referred to the maximum values of the 1 s regions containing the strongest signals in
the 5s sEMG signals collected. (The second paragraph of “2.3.2 Effectiveness evaluation of the system”)
7. There was no statistically significant difference analysis throughout the paper. Only the Cohen's
w values were shown. What would be the reason to select this small sample size in the current
study? Have the authors done any power analysis to determine the minimal sample size? ˆ
Response:Thank you very much for your valuable advice.
Relevant information has been added to “5.Limitations”. Details are as follows:
This paper is a case report of 5 BCI rehabilitation training instead of a randomized controlled study. The sample size is
relatively small, and the condition of each patient was different, so this report did not conduct paired sample t-tests of preand
post-intervention data, and only the Cohen's w values were shown. ( “5.Limitations”)
And the minimum sample size determined by the power analysis in this paper is 26.
8. Although the authors explained the implementation of both virtual and motor feedback in the
discussion section, the signals to generate the motor imagery or visual feedback were not clearly
explained. Also, these clarifications need to be included in the Materials and methods section. ˆ
12
Response:Thank you very much for your valuable advice.
Information related to the motor imagery or visual feedback has been included in the “Materials and methods section”
and “Discussion” . Details are as follows:
The system gave patients real-time visual and motor perception feedback, that is, during the training, the progress bar on
the right side of the display screen showed the intensity of the patient's motor imagery in real time (updates every 110ms);
when the intensity of the patient's motor imagery(i.e. Mscore) reached a certain threshold, the virtual character on the display
screen would start corresponding movements (swimming or cycling), and the virtual character's movement would generate
visual feedback; meanwhile, the rotary shafts of the system drove the patient's limb movement (update rate >1Hz), thus
giving the patient motor perception feedback [35,36]. (In the second paragraph of “2.2 L-B300 EEG Acquisition and
Rehabilitation Training System”)
Using this system requires patients to actively participate in the whole process and focus on their training done through
motor imagery. Mscore is the specific EEG signals used in this system to timely evaluate the degree of active target task
imagination of patients [45,46]. The Mscore was collected through the brain cap, decoded and transmitted to the terminal
devices (the rotary shafts and the display screen) via Bluetooth to control the rotary shafts and the virtual character on the
display screen. When Mscore meets with certain requirements, the rotary shafts can drive the patient's limb movement and
provide motor perception feedback to the patient's body sensation, while simultaneously giving visual feedback to the
patient on the display screen [36]. (In the first paragraph of “4. Discussion”)
9. Did the BCI system in the current study include FES during the rehabilitation training procedure?
If yes, please provide more details of the implementation of the FES system; if not, please clearly
specify the difference between the BCI system in the current study and other BCI system with
FES.
Response:Thank you very much for your valuable advice.
This system does not include FES.
Although the system does not combine FES, compared with other BCI systems, it still has certain advantages, that is, it
has dual feedback and bidirectional stimulation at the same time, and the system has short reaction time, timely feedback,
convenient and safe use, etc.
Because FES depends on the integrity and function of the neural reflex arc, the use of BCI system combined with FES
will be limited, for example, FES cannot play a role in the spinal cord shock phase caused by upper motor neuron injury,
severe lower motor neuron injury, or severe peripheral nerve injury. However, the BCI system used in this study is not
affected by the integrity and function of the neural reflex arc.
Whereas, FES also has the advantage of enhancing proprioceptive input by producing significant target muscle
contractions, which may improve the training effect.
In addition, the combination of BCI and FES to achieve voluntary motor function is one of the future research directions.